# Microbiological Characterisation of Community-Acquired Urinary Tract Infections in Bagamoyo, Tanzania: A Prospective Study

**DOI:** 10.3390/tropicalmed7060100

**Published:** 2022-06-12

**Authors:** Joseph Schmider, Nina Bühler, Hasina Mkwatta, Anna Lechleiter, Tarsis Mlaganile, Jürg Utzinger, Tutu Mzee, Theckla Kazimoto, Sören L. Becker

**Affiliations:** 1Institute of Medical Microbiology and Hygiene, Saarland University, 66421 Homburg, Germany; joseph.schmider@web.de (J.S.); nina.buehler@uks.eu (N.B.); anna.lechleiter@uks.eu (A.L.); tkazimoto@ihi.or.tz (T.K.); 2Ifakara Health Institute, Bagamoyo Branch, Bagamoyo P.O. Box 74, Tanzania; hmkwatta@ihi.or.tz (H.M.); tmlaganile@ihi.or.tz (T.M.); tmzee@ihi.or.tz (T.M.); 3Swiss Tropical and Public Health Institute, CH-4123 Allschwil, Switzerland; juerg.utzinger@swisstph.ch; 4University of Basel, CH-4003 Basel, Switzerland

**Keywords:** antimicrobial susceptibility, bacteria, diagnosis, *Escherichia coli*, infection, *Klebsiella* spp.

## Abstract

Urinary tract infections (UTIs) are among the most common infections in sub-Saharan Africa, but microbiological data to guide treatment decisions are limited. Hence, we investigated the bacterial aetiology and corresponding antimicrobial susceptibility patterns in outpatients with UTIs in Bagamoyo, Tanzania. Urine samples from symptomatic individuals were subjected to microbiological examinations for bacterial species identification using conventional methods and disc diffusion-based resistance testing. Subsequently, urine samples were transferred to Germany for confirmatory diagnostics using matrix-assisted laser desorption/ionization time-of-flight (MALDI-TOF) mass spectrometry and automated resistance testing. Overall, 104 out of 270 (38.5%) individuals had a positive urine culture and 119 putative pathogens were identified. The most frequently detected bacteria were *Escherichia coli* (23%), *Klebsiella* spp. (7%), *Enterobacter cloacae* complex (3%) and *Staphylococcus aureus* (2%). *E. coli* isolates showed high resistance against cotrimoxazole (76%), ampicillin (74%), piperacillin (74%) and fluoroquinolones (37%), but widespread susceptibility to meropenem (100%), fosfomycin (98%), piperacillin/tazobactam (97%) and amoxicillin/clavulanic acid (82%). The agreement between *E. coli* susceptibility testing results in Tanzania and Germany was ≥95%, except for piperacillin/tazobactam (89%) and ciprofloxacin (84%). Given the considerable resistance to frequently prescribed antibiotics, such as cotrimoxazole and fluoroquinolones, future research should explore the potential of oral alternatives (e.g., fosfomycin) for the treatment of UTIs in Tanzania.

## 1. Introduction

Urinary tract infections (UTIs) are among the most common community-acquired bacterial infectious diseases worldwide [1]. While many UTIs have a relatively mild course and can be treated effectively, repetitive infections with considerable long-term consequences, as well as acute complications are possible, e.g., ascending pyelonephritis or bloodstream infections [2]. Females are more often affected than males, and comorbidities (e.g., diabetes) are associated with higher odds of UTIs [3]. While Gram-negative bacteria, such as *Escherichia coli* are the most frequent causes of UTIs worldwide, the aetiology and underlying antimicrobial susceptibility patterns vary across settings [4]. In tropical and subtropical areas, the epidemiology of UTIs is poorly understood, due to the paucity of microbiological data, particularly in vulnerable populations, such as pregnant women or malnourished children [5]. As most individuals with UTIs present as outpatients to healthcare institutions, microbiological diagnostics are not routinely employed, and empirical antibiotic treatment is frequently prescribed. However, the adequacy and clinical efficacy of such treatments in light of regional resistance patterns are rarely assessed in low- and middle-income countries (LMICs) [6].

In many parts of sub-Saharan Africa there is a lack of adequate laboratory diagnostics, which is a barrier to the effective healthcare of infectious diseases [7]. In Tanzania, only a few studies on UTIs have been conducted, and these found that UTI was a common diagnosis in 18- to 50-year-old individuals presenting to an emergency department in Ifakara [8]. Of note, rather low susceptibility rates to commonly prescribed antibiotics, such as ampicillin and cotrimoxazole, were observed [9].

Against this background, our study aimed to (i) characterise the bacteria detected in urine samples from outpatients presenting with symptoms that were suggestive of community-acquired UTIs in the Bagamoyo area in Tanzania; and (ii) elucidate their antimicrobial susceptibility patterns, as determined by on-site disc diffusion testing in comparison to semi-automated testing in a European reference laboratory.

## 2. Materials and Methods

### 2.1. Ethics Statement

The study was part of a larger investigation entitled, ‘Profile of antibiotic-resistant bacteria from different sources’, for which ethical approval was obtained from the Tanzanian national ethics committee (reference: NIMR/HQ/R.8a/Vol. IX/2906). Written informed consent was sought from all participants before enrolment. For individuals aged below 18 years, informed consent was obtained from their legal guardians.

### 2.2. Study Area and Timing

The sample collection and primary laboratory testing were carried out from 16 October to 6 December 2019, in Bagamoyo, Pwani region, Tanzania. The Bagamoyo district has a population of approximately 310,000 inhabitants and covers an area of 9842 km^2^ of dry land along the Indian Ocean coast, north of Dar es Salaam, the economic capital of Tanzania. The climate is tropical with an average annual temperature of 28 °C. There are two rainy seasons: a minor one between November and February and a major one from March to May.

### 2.3. Study Design and Population

This prospective study was conducted in the Bagamoyo District Hospital, a referral hospital for the greater Bagamoyo area. Eligible individuals were all outpatients aged ≥2 years who sought medical care for symptoms suggestive of UTIs, such as dysuria, pollakiuria and lower abdominal pain. Exclusion criteria were acute or recent (previous 2 months) hospitalization, patients with an acute infectious disease diagnosis other than UTI, known pre-existing urological diseases (e.g., urinary incontinence) and individuals who were unable or unwilling to provide written informed consent. A standardised questionnaire was used to record basic patient characteristics, i.e., demographic data, the current symptomatology and potential risk factors for UTIs.

### 2.4. Laboratory Procedures

A clear midstream urine sample was obtained from each participant in the Bagamoyo District Hospital, adhering to detailed instructions provided by health care personnel. The collected samples were stored in a cooling box and transferred within 2 h to the laboratory of the Ifakara Health Institute (IHI) in Bagamoyo. Urine samples were cultured on three specific agar media, i.e., (i) 5% sheep blood agar (non-selective, universal agar medium); (ii) MacConkey agar (selective culture medium for Gram-negative bacteria); and (iii) UriSelect 4 agar (Bio-Rad Laboratories, Hercules, CA, USA; non-selective chromogenic medium to indicate the growth of different Gram-negative urinary pathogens). To detect the presence of any previous antibiotic therapy in urine samples, an inhibitor test was performed by spreading a *Bacillus subtilis* suspension on a Mueller–Hinton agar plate. Filter sheets were placed on this agar, and 10 μL of a urine sample was pipetted on each of them. A growth inhibition of *B. subtilis* indicated the presence of antibiotic substances in the respective samples.

All agar plate cultures were examined after an incubation of 24 and 48 h for signs of bacterial growth. Preliminary species identification was carried out using typical colony morphology on the different media, supplemented by Gram staining and simple biochemical tests (i.e., catalase, coagulase and oxidase). Antimicrobial susceptibility testing was performed by using the disc diffusion method on Mueller–Hinton agar, in accordance with recommendations put forth by the European Committee on Antimicrobial Susceptibility Testing (EUCAST, version 2019). The following antibiotic discs were used: ampicillin (disc content: 10 µg); ampicillin/sulbactam (10 µg/10 µg); amoxicillin/clavulanic acid (20 µg/10 µg); piperacillin (30 µg); piperacillin/tazobactam (30 µg/6 µg); cefuroxime (30 µg); cefotaxime (5 µg); ceftriaxone (30 µg); ceftazidime (10 µg); meropenem (10 µg); ciprofloxacin (5 µg); levofloxacin (5 µg); gentamicin (10 µg); cotrimoxazole (1.25 µg/23.75 µg); and fosfomycin (200 µg). If no clinical EUCAST breakpoints were available for a specific antibiotic, breakpoints developed by the Clinical & Laboratory Standards Institute (CLSI) were used. In addition to the microbiological tests, a urine dipstick was carried out on all samples to identify leucocyturia, nitrite and/or haematuria (DFI Co. Ltd.; Gimhae, Korea). All laboratory test results were communicated to the treating physician for further patient workup (e.g., targeted antibiotic treatment).

### 2.5. Confirmatory Microbiological Laboratory Testing

All culture-grown bacterial isolates were preserved in designated cryo-tubes and frozen at –80 °C. In August 2020, the frozen samples were transferred on dry ice to the Institute of Medical Microbiology and Hygiene, Saarland University in Homburg, Germany for confirmatory testing. Bacterial species identification was carried out using matrix-assisted laser desorption/ionisation time-of-flight (MALDI-TOF) mass spectrometry (Bruker Daltonics, Bremen, Germany). The MicroScan WalkAway 96 Plus system (Beckman Coulter, Brea, CA, USA) was used to perform microdilution-based resistance testing for most bacteria, except for streptococci, for which disc diffusion was employed. In the case of detected colistin resistance, confirmatory testing was carried out using broth microdilution, as suggested by EUCAST recommendations.

Antimicrobial susceptibility test results obtained in Tanzania and Germany were compared, and the categorical agreement was assessed [10], employing the test results in Germany as the reference standard. Very major errors (VME) were defined as susceptible results in Tanzania, whenever the reference test yielded a resistant pattern. Major errors (ME) were defined as resistant in Tanzania, while the reference test indicated a susceptible pattern. Minor errors (MinE) were defined as any condition in which either of the two tests showed an intermediate test result (which has recently been re-defined as “susceptible, increased exposure” by EUCAST) and the other one indicated a susceptible or resistant pattern.

### 2.6. Sample Size and Statistical Analysis

Based on previous epidemiological studies from Tanzania and elsewhere in sub-Saharan Africa, the urine culture positivity rate among symptomatic patients was estimated at 35–45%. Hence, we aimed to recruit approximately 240–290 symptomatic individuals to reach at least 100 culture-positive samples. For statistical analysis, data were double-entered into Microsoft Excel (version 2019, Microsoft Corporation, Redmond, WA, USA). Odds ratios (ORs) and confidence intervals (CIs) were calculated using the software RStudio (2022.02.0 Build 443, version R-4.1.2). A *p* value of <0.05 was considered as statistically significant.

## 3. Results

Over the 7 weeks of the study in late 2019, a total of 270 outpatients who presented with symptoms that were suggestive of UTIs in the Bagamoyo District Hospital were enrolled. There were more than 4 times more females than males (220 vs. 50). Patients ranged in age from 2 to 78 years with a mean age of 33.8 years (Table 1). The prevalence of self-reported symptoms that were suggestive of UTIs was 94% for dysuria, 77% for pollakiuria and 21% for flank pain. Potential urinary tract pathogens were detected in 104 patients, owing to a microbiological culture positivity of 38.5%. In one out of 10 cultures (10.4%), more than one potential pathogen was detected (21 cultures yielded two potential pathogens and seven cultures yielded three potential causative agents).

The prevalence of leucocyturia, haematuria and the presence of nitrite in the urine was 62.5%, 28.2% and 16.3%, respectively. All three conditions were significantly associated with urine culture positivity for bacteria (Table 2).

Among 119 potentially relevant bacteria, the majority was Gram-negative (89.9%). The most commonly detected pathogens in the cohort were *E. coli* (23%), *Klebsiella* spp. (7%) and *Enterobacter* spp. (3%). The most prevalent Gram-positive bacteria were *Staphylococcus aureus* (2%) and *Staphylococcus saprophyticus* (1%) (Table 3).

The antimicrobial susceptibility patterns of selected Gram-negative bacteria are displayed in Table 4. Except for ampicillin and piperacillin, resistance to most beta-lactams was relatively low among different pathogens. In *E. coli* isolates, resistance rates of 24% for second- and third-generation cephalosporin resistance, and 37% for fluoroquinolones were observed. Resistance to cotrimoxazole was also relatively frequent (15–76%). Of note, three out of seven Enterobacter strains displayed colistin resistance. Among the few detected *S. aureus* strains, none were methicillin-resistant.

The comparison between antimicrobial susceptibility testing that was carried out in Tanzania and Germany yielded concordant results for *E. coli* and *Klebsiella* spp., with a categorical agreement of ≥90% for most tested antibiotics. Of note, the categorical agreement was somewhat lower for ciprofloxacin in *E. coli* (84%), and the categorical agreement for most antibiotics was relatively low for *E. cloacae* complex strains (Table 5).

## 4. Discussion

In the current study pertaining to community-acquired UTIs in Bagamoyo in the eastern part of Tanzania, we found that 81.5% of all enrolled patients were female and that the vast majority of causative bacteria were Gram-negative, with *E. coli* (23%), *Klebsiella* spp. (7%) and *Enterobacter cloacae* complex (3%) being most frequently detected. Leucocyturia, haematuria and the presence of nitrite in urine samples were significantly associated with urine culture positivity, and there was a high concordance between antimicrobial susceptibility testing that was carried out on-site in Tanzania and later confirmatory testing in a reference laboratory in Germany, with the notable exception of the *E. cloacae* complex. We observed widespread resistance to specific antibiotics, such as ampicillin, cotrimoxazole and—especially in in *E. coli* isolates—to fluoroquinolones.

Our findings are in line with previous research that was carried out elsewhere in Tanzania. For example, a study from Moshi in the north-eastern part of the country identified *E. coli* and *Klebsiella* spp. as the two predominant species causing UTIs [9], while *E. coli* was also the major pathogen in a paediatric study from Mwanza in the northern part of Tanzania [11]. Likewise, resistance to ampicillin and cotrimoxazole was high, but only 11% of *E. coli* isolates displayed resistance to ciprofloxacin, which is lower than we observed in our study. Similar findings were reported from an analysis of UTIs in Tanzanian children with cerebral palsy [12], in people living with human immunodeficiency virus (HIV) infection [13] and in pregnant women [14]. Another prospective study from a northern area of the country identified the diagnosis of UTIs as being significantly associated with *E. coli* bloodstream infections, which underlines the possible severe consequences of untreated or inadequately treated infections [15]. *E. coli* is also the most frequently encountered pathogen in studies pertaining to UTIs outside Africa [16].

The relatively high resistance rate to the fluoroquinolone antibiotic ciprofloxacin is worrisome as this antibiotic is recommended as a first-line treatment in Tanzania’s official treatment guidelines for UTIs [17]. Indeed, previous research has demonstrated a clear link between widespread ciprofloxacin use in the community and a consecutive induction of resistance [18]. Such developments are further driven by the irrational use of oral antibiotics, which was previously reported in specific areas in Tanzania [19]. Besides the non-indicated use of antibiotics, the high rates of cotrimoxazole resistance might have been further spurred by the common use of this agent in HIV-infected individuals, e.g., to prevent *Pneumocystis jirovecii* pneumonia [20].

The polymyxin antibiotic colistin is commonly considered as a ‘last resort’ treatment option for the intravenous therapy of multi-drug resistant Gram-negative bacteria in humans. In 2015, plasmid-mediated genes that are able to confer resistance to colistin (e.g., *mcr-1*) were detected, which can rapidly spread between Gram-negative bacteria and pose a significant threat to human and animal health [21]. Colistin resistance has also been reported in Africa, and a potential link to its ongoing use in veterinary medicine is being discussed. In our study, three out of seven *E. cloacae* complex strains were phenotypically resistant to colistin. Even though we did not employ more advanced diagnostics for *mcr-1*, this is a worrisome finding, as colistin resistance was previously considered rare. There is a paucity of data on the prevalence of colistin resistance in Gram-negative pathogens in Tanzania, and yet, a previous study from the island of Zanzibar found unexpectedly high rates among hotel employees, e.g., an *mcr-1* positivity in *E. coli* strains of 55% [22]. Further research is urgently needed to quantify the magnitude of this phenomenon in clinical isolates in Tanzania and elsewhere in sub-Saharan Africa.

Our study has several limitations that are offered for consideration. First, our findings (e.g., regarding the colistin resistance of *E. cloacae* complex) need to be confirmed in larger studies and conducted over a period of at least 1 year to address seasonal issues, ideally employing a multicentric design. Such investigations should also try to elucidate the resistance mechanism giving rise to specific phenotypic patterns (e.g., AmpC beta-lactamases, extended-spectrum beta-lactamases). Second, while our report elucidates the microbiological aetiology of community-acquired UTIs, specific data on risk factors and the general prevalence in the population, as well as on co-infections are warranted. Third, some fastidious or slowly growing pathogens such as *Aerococcus urinae* or *Corynebacterium jeikeium* might not have been detected by our diagnostic approach, and hence, the actual number of culture-positive samples might have been higher. Fourth, in future studies, it will be interesting to test the antimicrobial susceptibility patterns of additional, potentially promising oral antibiotics, such as nitrofurantoin and pivmecillinam, which are nowadays frequently used in Europe and elsewhere [18]. Fifth, a more in-depth molecular characterization will help to elucidate the potential clonal spreads of specific bacterial strains in the study area. Sixth, for comparative antimicrobial susceptibility testing across different sites, identical antibiotics should be employed, which was not always the case in our study (e.g., for the assessment of third-generation cephalosporins, ceftriaxone was used in Tanzania, while cefotaxime was used in Germany).

## 5. Conclusions

We report on the bacterial aetiology and resistance profile of community-acquired UTIs in Bagamoyo, Tanzania. We found considerable resistance to certain beta-lactam antibiotics, fluoroquinolones, and cotrimoxazole, as well as low, but notable rates of colistin resistance. Our findings call for future studies to explore the therapeutic potential of alternative oral antibiotics, such as fosfomycin, for the treatment of UTIs in Tanzania and elsewhere in sub-Saharan Africa.

## Figures and Tables

**Table 1 tropicalmed-07-00100-t001:** Baseline epidemiological characteristics of a study on community-acquired urinary tract infections (UTIs) among 270 individuals in Bagamoyo, Tanzania, in late 2019.

Age (Years)	Total	Sex		UTI Pathogen Detected	
		Male	Female	Male	Female
<18	38	14	24	1 (7%)	10 (42%)
18–35	135	12	123	2 (17%)	44 (36%)
36–59	64	15	49	3 (20%)	25 (51%)
≥60	33	9	24	2 (22%)	17 (71%)
Total	270	50	220	8 (16%)	96 (44%)

**Table 2 tropicalmed-07-00100-t002:** Association between leucocyturia, haematuria and the presence of nitrite in the urine with urine culture positivity for bacterial growth among 270 individuals with symptoms suggestive of urinary tract infections (UTIs) in Bagamoyo, Tanzania, in late 2019.

Variable		Urine Culture Positivity (n = 104)	Odds Ratio (95% Confidence Interval)	*p*
**Leucocyturia**				
Positive	169	98		
Negative	101	6	21.6 (8.8–63.8)	<0.001
**Haematuria**				
Positive	76	41		
Negative	194	63	2.4 (1.4–4.3)	0.001
**Nitrite**				
Positive	44	42		
Negative	226	62	54.8 (13.6–480.1)	<0.001

**Table 3 tropicalmed-07-00100-t003:** Prevalence and species distribution of causative bacterial pathogens in a cohort of 270 individuals with clinical symptoms suggestive of community-acquired urinary tract infections (UTIs) in Bagamoyo, Tanzania in late 2019.

Uropathogenic Bacteria	Prevalence	
	n	%
**Gram-negative Bacteria**	107	40
*Escherichia coli*	62	23
*Klebsiella* spp. ^1^	18	7
*Enterobacter cloacae* complex	7	3
*Acinetobacter junii*	5	2
*Pseudomonas* spp. ^2^	4	2
*Proteus mirabilis*	4	2
Other species ^3^	7	3
**Gram-positive Bacteria**	12	4
*Staphylococcus aureus*	6	2
*Staphylococcus saprophyticus*	3	1
*Streptococcus agalactiae*	2	1
*Enterococcus faecalis*	1	0.4%

^1^ *Klebsiella pneumoniae* and *Klebsiella oxytoca*; ^2^ *Pseudomonas aeruginosa* and *Pseudomonas stutzeri*; ^3^ *Citrobacter freundii*, *Morganella morganii*, *Pantoea antophila*, *Pantoea stewartii*, *Providencia stuartii* and *Comamonas kerstersii.*

**Table 4 tropicalmed-07-00100-t004:** Antimicrobial susceptibility patterns of the three most commonly isolated Gram-negative bacteria in a study among 270 individuals from Bagamoyo, Tanzania with community-acquired urinary tract infections (UTIs) in late 2019.

Antibiotic	*Escherichia coli* (n = 62)		*Klebsiella* spp. (n = 18)		*Enterobacter cloacae* Complex (n = 7)	
	Resistant		Resistant		Resistant	
	n	%	n	%	n	%
Ampicillin ^1^	46	74	18	100	5	71
Amoxicillin/clavulanic acid	46	18	1	6	6	86
Piperacillin ^1^	46	74	18	100	3	43
Piperacillin/tazobactam	2	3	0	0	0	0
Cefuroxime	15	24	1	6	6	86
Cefotaxime	15	24	1	6	0	0
Ceftazidime	9	15	1	6	0	0
Meropenem	0	0	0	0	0	0
Ciprofloxacin	23	37	0	0	0	0
Levofloxacin	23	37	0	0	0	0
Gentamicin	12	20	0	0	0	0
Cotrimoxazole	47	76	6	33	1	15
Colistin	0	0	0	0	3	43
Fosfomycin ^2^	1	2				

^1^ *Klebsiella* spp. are naturally resistant to ampicillin and piperacillin. ^2^ According to EUCAST, clinical breakpoints for oral fosfomycin are only applicable to *E. coli*.

**Table 5 tropicalmed-07-00100-t005:** Comparison of antimicrobial susceptibility testing results of pathogens detected from patients with urinary tract infections (UTIs) in Bagamoyo, Tanzania. Susceptibility testing was carried out in Tanzania (using agar disc diffusion) and in Germany (using MicroScan WalkAway).

***Escherichia coli* (n = 61 ^1^)**	**c.a.**	**%**	**MinE**	**%**	**ME**	**%**	**VME**	**%**
Cefotaxime ^2^	59	97	0	0	1	2	1	2
Cefuroxime	60	98	0	0	0	0	1	2
Ciprofloxacin	51	84	5	8	3	5	2	3
Fosfomycin	59	97	0	0	1	2	1	2
Piperacillin/tazobactam	54	89	5	8	2	3	0	0
Cotrimoxazole	60	98	0	0	0	0	1	2
Amoxicillin/clavulanic acid ^3^	58	95	0	0	2	3	1	2
***Klebsiella* spp. (n = 18)**	**c.a.**	**%**	**MinE**	**%**	**ME**	**%**	**VME**	**%**
Cefotaxime ^2^	18	100	0	0	0	0	0	0
Cefuroxime	18	100	0	0	0	0	0	0
Ciprofloxacin	18	100	0	0	0	0	0	0
Piperacillin/tazobactam	17	94	0	0	1	6	0	0
Cotrimoxazole	17	94	0	0	0	0	1	6
Amoxicillin/clavulanic acid ^3^	18	100	0	0	0	0	0	0
** *Enterobacter cloacae* ** **Complex (n = 7)**	**c.a.**	**%**	**MinE**	**%**	**ME**	**%**	**VME**	**%**
Cefotaxime ^2^	7	100	0	0	0	0	0	0
Cefuroxime	0	0	1	14	0	0	6	86
Ciprofloxacin	5	71	2	29	0	0	0	0
Piperacillin/tazobactam	7	100	0	0	0	0	0	0
Cotrimoxazole	5	71	0	0	1	14	1	14
Amoxicillin/clavulanic acid ^3^	2	29	0	0	0	0	5	71

Abbreviations: c.a., categorial agreement; ME, major error; MinE, minor error; VME, very major error. ^1^ 61 out of 62 detected *E. coli* strains were subjected to resistance testing in Tanzania. ^2^ Results based on a comparison between cefotaxime (tested in Germany) and ceftriaxone (tested in Tanzania). ^3^ Results based on a comparison between amoxicillin/clavulanic acid (tested in Germany) and ampicillin/sulbactam (tested in Tanzania).

## Data Availability

Data are available on request from the corresponding author.

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
