# Peer review of "Microbiological Characterisation of Community-Acquired Urinary Tract Infections in Bagamoyo, Tanzania: A Prospective Study"

_tropicalmed, 2022, doi:10.3390/tropicalmed7060100_

Round 1
Reviewer 1 Report
The aim of the study was to investigated the bacterial aetiology and corresponding antimicrobial susceptibility patterns in outpatients with UTIs in Bagamoyo Distric Hospital, Tanzania. The study results might be interesting, however in the local area mostly.
My major concern is:
- What about the antibiotics resistance mechanisms that are routinely checked? Were any of them detected in the group of pathogens isolated?
- Lines 191-192, in my opinion this comparison is not correct. The activitis of the two antimicrobials combinations are different.
My minor concerns are:
- Keywords should be listed in an alphabetical order, in my opinion.
- Line 33 – unnecessary “)”.
- There is a constant lack of manufacturers for the used reagents, equipment.
- Line 120-121– it should be explained why “ In case of detected colistin resistance, confirmatory testing was carried out using microdilution”.
- Italics is missing in some places.
- Table 4 – natural resistance should be mentioned/explained.
- Fonts are different in the manuscript, e.g. Table 5.
- Antibiotics disc - it should be specify what is the exact concentration of each antimicrobial.
- It would be better to give the % numbers with one decimal place, e.g. in Tables.
- Line 188 – why did you use the word “only”?
- Lines 189-192 – this fact should be mentioned/discussed in the Discussion section.
- It should be discussed in the Discussion section whether the appearing unusual antimicrobial resistance phenotypes may result from clonal spread of the bacteria, also in the group of outpatients.
However, all the points mentioned above do not decrease the overall value of the research.
Author Response
Please see the detailed point-by-point response, which we have submitted as individual file.

Reviewer 2 Report
The authors describe an interesting article regarding the microbiological characterization of community-acquired urinary tract infections. Introduction and materials and methods are clearly described. Resistance to amoxicillin and sulfamethoxazole-trimethoprim are expected.
The results of Table 2 can be put in the text because the results are already known.
Colistin resistance from table 4 cannot be considered as reliable as there are only 7 patients included in the analysis (3/7). The authors should comment on this conclusion as a limitation of the study.
Table 5 is not clearly written, it probably means samples analyzed in Germany. That should be written more clearly.
The best part of the article is an analysis of irrational use and government recommended use of fluoroquinolones. It serves as a warning to all of us. The discussion is correct.
Author Response

(The authors gave the same response as above.)

Reviewer 3 Report
Congratulation on the present study!
UTI and antibiotic resistance represent one of the biggest threats to the modern world.
I would have some recommendations.
1. In the MaterialS and Methods section you can include more details about the studied population. For example: are there any chronic urological affections like surgical past or urinary incontinence?
2. The discussion section can be extended. For example, you can compare your data with other European studies. Extensive studies were conducted in Germany, Poland, Spain, Greece, Hungary. Petca et al. conducted an extensive research in 2019 on the general Romanian population https://doi.org/10.31925/farmacia.2019.6.9
Author Response

(The authors gave the same response as above.)

Round 2
Reviewer 2 Report
The authors have sufficiently improved the manuscript to warrant publication.